# Identification of Hub Genes and Immune Infiltration in Pediatric Biliary Atresia by Comprehensive Bioinformatics Analysis

**DOI:** 10.3390/children9050697

**Published:** 2022-05-10

**Authors:** Yajing Li, Huichu Ye, Yingxue Ding

**Affiliations:** Department of Pediatrics, Beijing Friendship Hospital, Capital Medical University, Beijing 100050, China; yajing@mail.ccmu.edu.cn (Y.L.); yhc2022@mail.ccmu.edu.cn (H.Y.)

**Keywords:** biliary atresia, children, WGCNA, immune-related genes, biomarkers

## Abstract

Background: Biliary atresia (BA) is the leading cause of pediatric liver failure and pediatric liver transplantation worldwide. Evidence suggests that the immune system plays a central role in the pathogenesis of BA. Methods: In this work, the novel immune-related genes between BA and normal samples were investigated based on weighted gene co-expression network analysis (WGCNA) and the deconvolution algorithm of CIBERSORT. Results: Specifically, 650 DEGs were identified between the BA and normal groups. The blue module was the most positively correlated with BA containing 3274 genes. Totally, 610 overlapping BA-related genes of DEGs and WGCNA were further used to identify IRGs. Three IRGs including VCAM1, HLA-DRA, and CD74 were finally identified as the candidate biomarkers. Particularly, the CD74 biomarker was discovered for the first as a potential immune biomarker for BA. Conclusions: Possibly, these 3 IRGs might serve as candidate biomarkers and guide the individualized treatment strategies for BA patients. Our results would provide great insights for a deeper understanding of both the occurrence and the treatment of BA.

## 1. Introduction

Biliary atresia (BA) is one of the most severe hepatobiliary diseases of early infancy with a prevalence of 1/19,000–1/5000 varied in different countries [1,2,3,4]. Typically, BA is characterized by an inflammatory and fibrotic obliteration of the extrahepatic bile ducts, leading to cholestasis and irreversible liver failure. Without any treatment, BA would lead to death from end-stage liver disease in the first two years of life [5]. To date, BA remains the leading cause of pediatric liver transplantation all over the world (32%) [6]. Therefore, both the early screen technology and the accurate diagnosis for BA in the early stage are urgently required for pediatrics. 

The pathogenesis of BA is greatly complex and unclear to date. Currently, theories of pathogenesis were proposed to include defects in embryogenesis, gene sequence factors, toxins, different viral infections, aberrant expression of immune cells, and immune regulatory factors and susceptibility factors [5,7]. Among all the factors linked to the pathogenesis of BA, the immune system is recognized as the core factor, which has been proved by the infiltration of the liver by inflammatory cells and the overexpression of cytokines and/or chemokines at the time of diagnosis [7,8,9,10]. We presume that the genes especially immune-related genes (IRGs) might play a vital role in the occurrence of BA and serve as biomarkers for this disease. Nevertheless, current studies mostly focused on the immune cell levels rather than the gene-related research. Accordingly, it would be of great interest to perform the corresponding gene level studies to identify the role of IRGs in the occurrence of BA. 

WGCNA (weighted gene co-expression network analysis) is a significantly useful method that divides gene co-expression networks of complex biological processes into several characteristic modules and analyzes their association with clinical traits to reveal the specific association between genes and clinical features [11]. In previous studies, WGCNA has been widely used in hepatobiliary diseases. CIBERSORT is a computational tool to estimate the abundances of member cell types in a mixed cell population [12] which attracted great attention in studying cell heterogeneity. We expect that these computational analytical methods of WGCNA and CIBERSORT would be significantly useful to explore the diagnosis of genes and therapeutic biomarkers in the clinical practice of BA.

In this work, BA gene expression data were downloaded from the NCBI Gene Expression Omnibus (GEO) database (https://www.ncbi.nlm.nih.gov/geo (accessed on 10 November 2021)). WGCNA was used to construct key modules and CIBERSORT was applied to analyze the immune infiltration in BA. Moreover, we further explored and verified the novel immune-related genes, which might serve as potential biomarkers for diagnosis and guide the personalized strategies for treating BA patients. Our computational analytical results might provide great insights for understanding the occurrence of BA. 

## 2. Materials and Methods

Data download and processing: GSE122340 and GSE46960 datasets were downloaded from the NCBI GEO database. A total of specimens of 7 healthy controls and 171 infants with BA were collected from dataset GSE122340. The detection platforms were based on GPL16791. The dataset GSE46960 was based on the GPL6244 platform, containing 7 liver samples from healthy controls and 64 liver samples from BA patients. Moreover, GSE122340 was employed as a training dataset and GSE46960 was used as the independent validation cohort.

Identification and functional enrichment analysis of differentially expressed genes (DEGs): DEGs were analyzed using the limma package in R (version 4.12, University of Auckland, New Zealand). with the cutoff criteria of adjusted *p*-value < 0.05 and |log2 foldchange| > 1 [13]. Volcano plots and heatmaps were created using the ggplot2 package in RTo explore the biological functions of DEGs, GO and KEGG pathway analyses were conducted through the cluster Profiler package [14]. Specifically, *p* < 0.05 was considered statistically significant.

Identification of key modules using WGCNA: WGCNA package in R was used to construct the weighted gene co-expression network based on all the genes involved in the training cohort [11]. WGCNA can detect highly relevant genes and aggregate them into the same co-expression modules associated with clinical traits. Soft powers (β = 4) were applied to the dataset GSE122340 to construct the scale-free networks. Then, the hierarchical clustering dendrogram was built to aggregate genes with similar expressions into the same co-expression module. Then the module-trait relations between modules and clinical traits were explored for functional modules from the co-expression network. Finally, the module with the largest correlation coefficient was considered to be the potential module which was mostly correlated to clinical traits. The corresponding module with the highest correlation coefficient was utilized in the next analysis.

Analysis of immune infiltration using the CIBERSOFT algorithm: Immune cell infiltration in BA was evaluated using CIBERSORT (https://cibersort.stanford.edu/ (accessed on 20 January 2019)) which is a newly developed and superior tool for calculating the cell proportion of complex tissues based on gene expression profiles [12]. The LM22 gene file of CIBERSORT was used to define 22 immune cell subsets. During the data analysis, the results were filtered according to *p*-value < 0.05, and the immune cell composition of each sample was shown in the bar plot.

Identification and verification of hub immune-related genes: Considering the critical role of the immune system in the pathogenesis of BA, IRGs may serve as biomarkers for BA. After integrating the results of WGCNA and DEGs, the potential hub genes were obtained. Then, IRGs were downloaded from the import shared database (http://www.immport.org/ (accessed on 24 March 2020)), which were subsequently overlapped with the potential hub genes to indicate the candidate IRGs for BA. The receiver operator characteristic curve was plotted using the pROC package [15] in R.Subsequently, the area under the curve was utilized to determine the hub genes which might discriminate BA from normal samples. Finally, GSE46960 including 64 BA samples was used as a validation cohort to further confirm the real hub IRGs.

PPI network construction and gene correlation analysis: To further investigate the connection between candidate hub genes and other BA-related IRGs at the protein level, the Search Tool for the Retrieval of Interacting Genes (STRING, https://string-db.org/ (accessed on 18 November 2021)) was used to construct the PPI network [16]. During the analysis, the lowest interaction score was set as 0.4, and any isolated node in the network was removed. Finally, the association between any two hub genes was capered based on the corrplot package in R.

## 3. Results

### 3.1. Identification of DEGs

In the data set of GSE122340 as the training cohort, 650 DEGs were identified between the BA and normal. These DEGs were determined according to the criterion with *p*-values < 0.05 and |log2FoldCharge| > 1. Among these DEGs, 504 genes were up regulated and 146 genes were downregulated. The volcano plot of all DEGs and heatmap of the top 50 DEGs are shown in Figure 1A,B. Under the same criteria, 485 DEGs were discovered in GSE46960.

### 3.2. Functional and Pathway Enrichment Analysis of DEGs

To further explore the function of DEGs associated with BA, the pathway enrichment analysis of GO and KEGG was carried out using the Bioconductor package clusterProfiler in R. In GSE122340, the GO enrichment analysis revealed that for BP, the BA-related genes were mainly related to RNA splicing, nucleocytoplasmic transportation, and response to a toxic substance. For CC, the BA-related genes were mainly involved in nuclear speck, collagen-containing extracellular matrix, and blood microparticle. As for MF, the BA-related genes were mainly relevant in tetrapyrrole binding, antioxidant activity, and heme-binding (Figure 2A,B). As shown in Figure 2C,D, BA-related genes were mainly enriched in diabetic cardiomyopathy, spliceosome, and chemical carcinogens DNA adducts pathways.

### 3.3. Co-Expression Network Construction and Key Modules Visualization 

Sample clustering was performed based on Pearson’s correlation matrices using the average linkage method. No outliers were detected during the sample clustering (see Figure 3A). When the correlation coefficient threshold was set as 0.9, the soft-thresholding power was selected as four in Figure 3B. The correlation between genes and modules, and between different genes were also analyzed. The modules with more genes clustered were blue, purple, and black (Figure 3C,D). Subsequently, the relationship between modules and traits was determined (see Figure 3E). A total of 7 gene modules were obtained. The correlation between the blue module and the clinical phenotype was the highest (Pearson correlation ratio: 0.31, *p* < 0.001) which suggested that the genes in the blue module were significantly associated with the clinical phenotype of BA. Figure 3F shows the scatter plot of gene significance in the blue module. It reveals that the blue module from WGCNA has a significant correlation with gene significance of BA (correlation coefficient:0.78, *p* < 1 × 10^−200^).

### 3.4. Immune Cell Profiling and Microenviroment Analysis of BA

Figure 4A summarizes the distribution of various immune cells in each sample in the GSE122340 cohort. Different colors represented various types of immune cells. The height of each color represented the percentage in the sample, and the sum of the percentage of various immune cells was 1. These data demonstrated that the main infiltrating cells include M2 macrophages, CD8+ T cells, naïve CD4+ T cells, memory resting CD4+ T cells, memory activated CD4+ T cells, follicular helper T cells, regulatory T cells, naïve B cells, and memory B cells. As shown in Figure 4B, the proportions of different infiltrated immune cell subpopulations were weakly or moderately correlated. For instance, there was a positivecorrelation between CD8+ T cells and M1 macrophage with the correlation coefficient of 0.7while NK cell resting and NK cell activated are negatively correlated with the correlation coefficient of −0.64. As compared with normal samples, BA samples generally contained a higher proportion for the memory B cells (*p* = 0.035), gamma delta T cells (*p* = 0.035), and the resting dendritic cells (*p* = 0.044) (see Figure 4C).

### 3.5. Identification of Significant IRGs for BA

Totally, 610 potential hub genes were overlapped with the immune-related genes downloaded from the online database, which were obtained by integrating the results of WGCNA and DEGs (see Figure 5A). Specifically, 30 differential IRGs were obtained (Figure 5B). By integrating with the differential IRGs of the validation cohort (Figure 5C), 3 IRGs were finally identified for further analysis (Figure 5D). To further confirm the hub IRGs to effectively distinguish BA and normal samples, ROC curves of the 3 genes were plotted. These results revealed that all 3 IRGs could distinguish BA samples from normal samples with AUC > 0.7 (Figure 6A,C,E). Moreover, 3 IRGs also showed significant differences in the validation cohort GSE46960 (Figure 6B,D,F). Thus, VCAM1, HLA-DRA and CD74 were finally identified as the candidate biomarkers of BA.

### 3.6. PPI Network Construction and Gene Correlation Analysis

The PPI network for all 30 differential IRGs was constructed using the STRING tool. The corresponding result of the PPI network demonstrated that all 3 potential biomarkers had relationships with other proteins (Figure 7A). Therefore, these 3 potential gene changes cause coding protein changes and lead to related protein changes which might induce the occurrence of BA. Moreover, the interaction of the expression of the 3 hub genes was calculated and CD74 was proven to have a strong positive correlation with HLD-DRA, suggesting that the 3 hub genes might interact with each other (Figure 7B).

## 4. Discussion

To date, BA is widely considered the leading cause of both pediatric liver failure and liver transplantation worldwide [6,17,18]. Early screen technology and the accurate diagnosis for BA in the early stage are urgently required for pediatrics. The immune system is recognized as the core factor for BA. In recent years, biomarkers in the molecular levels have attracted the most attention. Hirotaka demonstrated a high concentration of MMP-7 in BA and might serve as a useful marker for diagnosis [19,20]. Based on CIBERSOFT, Zhang explored the microenvironment of BA and found that CXCL8 might serve as the hub gene of BA with a small sample size [21]. In this work, comprehensive bioinformatics was utilized to explore immune-related genes and microenvironment with a more updated public database including more cases and more consistent clinical features. Three immune-related genes were finally identified as the candidate biomarkers and a CD74 biomarker was discovered for the first as a potential screening biomarker for BA.

VCAM-1 (vascular cell adhesion molecule-1), a cell adhesion molecule belonging to the immunoglobulin supergene family, was identified as one of the immune biomarkers of BA in this study. VCAM-1 is predominantly expressed on the surface of endothelial cells and is critical to the adhesion of lymphocytes to target cells [22]. In our work, VCAM-1 levels were significantly upregulated in the liver in BA patients compared with normal patients. A prior clinical study demonstrated higher serum VCAM-1 levels in BA patients than in normal controls. Moreover, the higher the degree of liver fibrosis in BA, the more significant the difference in VCAM-1 expression levels [23]. Meanwhile, it was also expressed in the hepatic parenchyma including the portal tract and the portal vein in BA patients, especially with advanced cirrhosis which suggested that ongoing cirrhosis could be mediated by VCAM-1 through humoral and cell-mediated immune interaction [24]. Therefore, VCAM-1 might play a key role in the occurrence and liver fibrosis progression and could be used as a screening biomarker and prognosis predictor in BA patients. 

HLA-DRA (human leukocyte antigen DR alpha chain) is a homo sapiens major histocompatibility complex (MHC) class II antigen which might play an important role in immunity by presenting peptides derived from extracellular proteins [25]. A previous study showed that HLA-DR was expressed on hepatocytes and bile duct cells in BA patients but was not present in control patients [26,27]. Subsequently, researchers found that it was also abnormally expressed in microvilli of the bile duct in patients with BA and was inversely associated with the short-term outcome after operation [27]. It is indicated that aberrant HLA-DR expression may play a pathogenic role in cirrhosis progression in BA. Recently, the role of HLA-DRA has attracted great attention in the diagnosis of tumors [25,28]. This study found that the expression of HLA-DRA in the BA liver was aberrantly upregulated which further indicates that HLA-DRA might play a pathological role in BA.

CD74 (MHC class II invariant chain) is a non-polymorphic type II transmembrane protein. Initially, CD74 was thought to be an MHC class II chaperone expressed on classical antigen-presenting cells (APCs), such as dendritic cells and macrophages [29]. However, more evidence supports that CD74 has many more biological functions in different situations. In recent years, CD74 has been recognized as a high-affinity cell membrane receptor for macrophage migration inhibitory factor (MIF), contributing to the activation of the ERK (extracellular signal-regulated kinase) pathway and Akt pathway [30,31]. Furthermore, the role of MIF and its cell membrane receptor CD74 is mainly focused on the protection against injury and healing promotion in different parts of the body including the kidney, lung, gut, heart, and nervous system [31,32,33,34,35,36]. Therefore, activation of the MIF-CD74 pathway seems to mainly correlate positively to protection against injury and healing promotion. However, elevated levels of CD74 and MIF positively correlated with worsening inflammation of lupus nephritis from the proinflammatory effects of MIF/CD74 signaling [30]. Our study revealed that CD74 is significantly upregulated in BA compared with normal samples. Moreover, this study is the first to identify the specific relationship between CD74 and the occurrence of BA. However, it is unclear how CD74 impacts the progression of BA. We propose that the MIF-CD74 pathway might play a significant role in BA which would be an attractive and interesting area for future research. Furthermore, a strong correlation between CD74 and HLA-DRA was obtained at the transcriptional level. Hence, we assumed that these three potential hub genes might affect each other in the occurrence of BA but the specific mechanism needs to be clarified in future investigation.

We applied CIBERSORT to explore the microenvironment and correlation between immune cell infiltration and BA. The immune cells differed significantly including memory B cells, gamma delta T cells, and resting dendritic cells. B memory cells were rapidly reactivated to produce antibodies and establish a second line of defense [37]. In the rhesus rotavirus (RRV)-induced neonatal mouse model of BA (murine BA), B cell-deficient mice are protected from developing BA [38]. However, if B cells were transferred into RRV-infected, B cell-deficient mice, T cells, and macrophages would be reinstated and start biliary injury again [39]. The latest study published in CELL revealed that hepatic B cell lymphopoiesis remained intact after birth, which may also regulate the functions of myeloid cells, T cells, and NK cells through MHC-mediated antigen presentation or soluble mediators apart from producing autoantibodies. Through the depletion of B cells by an anti-CD20 antibody, symptoms significantly improved compared with untreated RRV mice [40].

Gamma delta T cells are a kind of unique and conserved population of lymphocytes that have attracted great interest in recent years due to their critical roles in many types of immune response and immunopathology [41]. Recently, the ability of gamma delta T cells to produce large amounts of IL17 has been explored [42,43]. The role of IL17-positive gamma delta T cells has been evaluated in various models of infection and autoimmunity. Similarly, gamma delta T cells were upregulated in liver samples of both BA patients and murine models compared with healthy controls [44]. With suppression of the IL-17 production of gamma delta T cells by AM80, hepatic inflammation is ameliorated in mice suffering from BA [45]. It is suggested that gamma delta T cells can produce IL17 and might contribute to the inflammation and destruction of the extrahepatic and intrahepatic bile ducts.

Dendritic cells are a family of relatively rare immune cell subtypes that establish unique cellular networks and produce distinct cytokine signals with complementary functions in antigen sensing and activation of T lymphocytes [46]. Dendritic cells were reported to exist before developing any symptoms of BA and localize to bile ducts after RRV infection which then modify T cell and natural killer cell activation and epithelial injury in experimental BA [47]. Our study showed that the number of resting dendritic cells in the BA patients was statistically lower than that in the healthy controls, while the activated dendritic cells were upregulated compared with the healthy controls, suggesting that dendritic cells are important in the development of BA and deserve further investigation.

This study has several limitations. A larger sample size is needed in future studies. We should further validate the discriminative ability of the candidate biomarker genes in animal models and in our own cohort. Finally, functional and mechanism studies of the hub genes identified here are needed. 

## 5. Conclusions

In summary, the novel immune-related genes and microenvironment between BA and normal liver were investigated based on WGCNA and CIBERSORT. Three IRGs including VCAM1, HLA-DRA, and CD74 were finally identified as the candidate biomarkers. Particularly, the CD74 biomarker was discovered for the first time as a potential immune biomarker for BA. Possibly, these three IRGs might serve as candidate biomarkers and guide the individualized treatment strategies for BA patients. CIBERSOFT analysis demonstrated that three immune cell types might contribute to the pathogenesis of BA including memory B cells, gamma delta T cells, and resting dendritic cells. Our results would provide great insight for a deeper understanding of the occurrence and treatment of BA. 

## Figures and Tables

**Figure 1 children-09-00697-f001:**
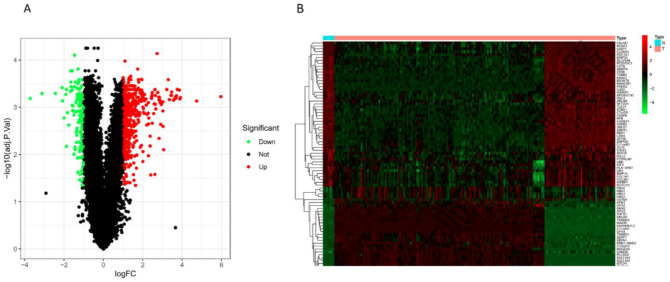
Visualization of differentially expressed genes (DEGs). (**A**) Volcano map of the DEGs in the training cohort. (**B**) Heatmap of the top 50 upregulated and downregulated genes.

**Figure 2 children-09-00697-f002:**
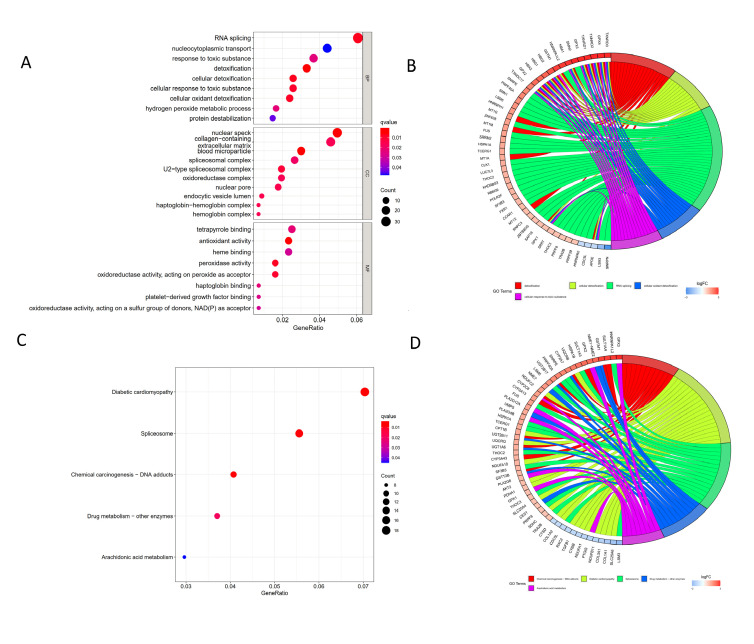
Functional and pathway analysis through GO and KEGG. (**A**) MF, BP, and CC in GO categories of DEGs. (**B**) Circos plot indicating the relationship between genes and GO terms. (**C**) KEGG analysis of DEGs. (**D**) Circos plot indicating the relationship between KEGG pathways.

**Figure 3 children-09-00697-f003:**
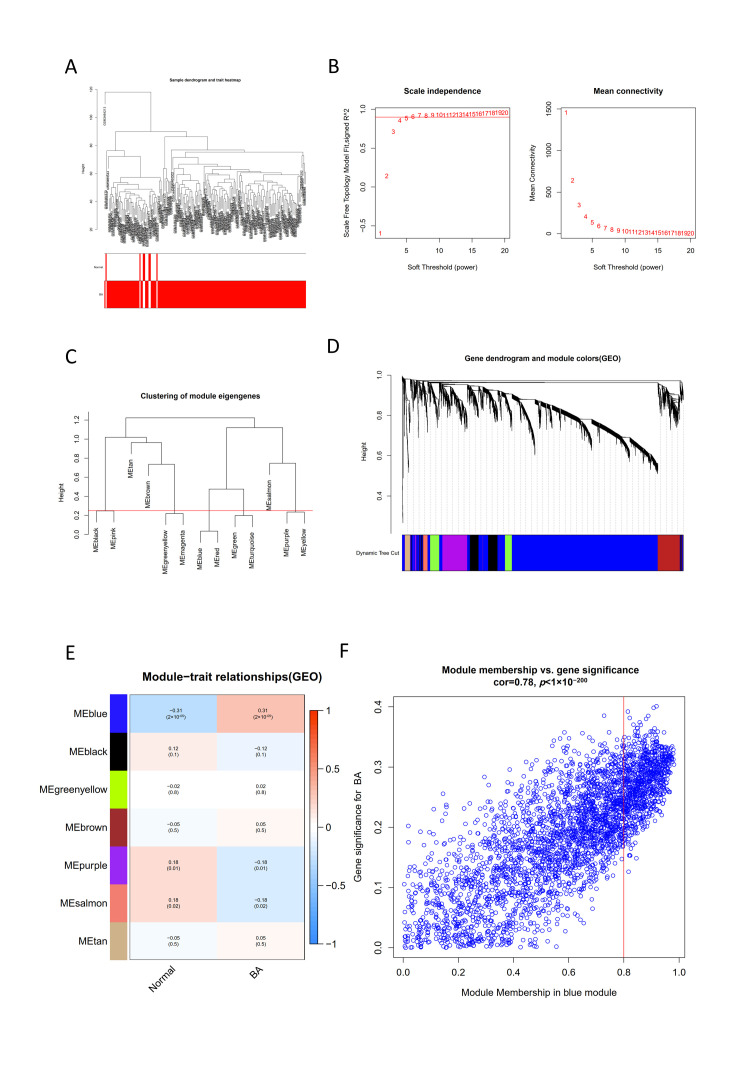
Correlated key module construction with BA through WGCNA. (**A**) Sample clustering dendrogram of the GSE122340. (**B**) Clustering dendrograms of genes. Analysis of the scale-free fit index (**left**) and the mean connectivity (**right**) for various soft-thresholding powers. (**C**) Dendrogram of all differentially expressed genes (DEGs) based on a dissimilarity measure. (**D**) An eigengene dendrogram identified groups of correlated modules. (**E**) Module–trait relationships between genes and clinical traits of BA. Each cell contains the correlation coefficient and *p*-value. (**F**) Scatter plot of gene significance in the blue module.

**Figure 4 children-09-00697-f004:**
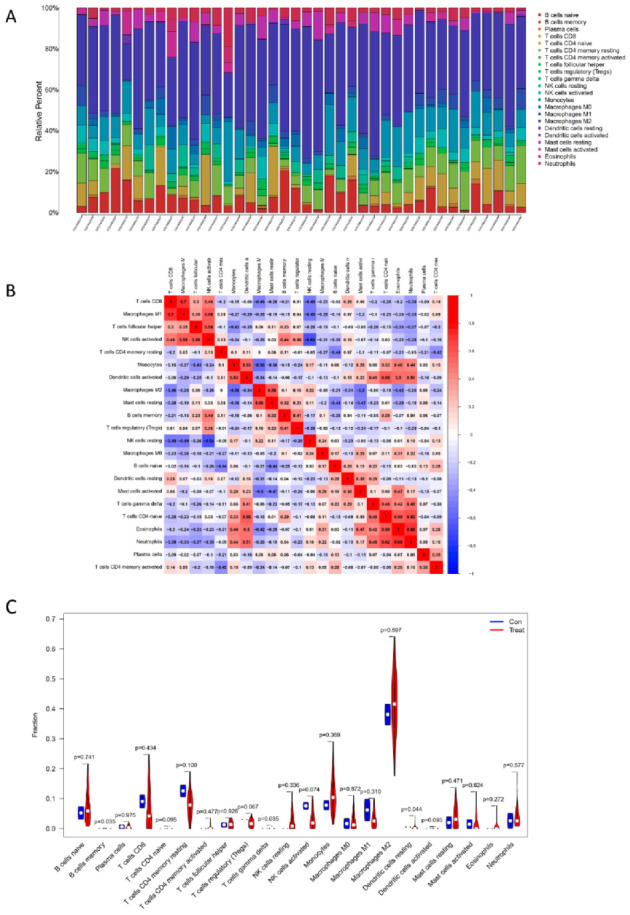
The fraction of the immune cells in the BA and control groups. (**A**) The bar chart indicates the distribution of the 22 subsets of immune cells. X-axis: each GEO sample; y-axis: percentage of each kind of immune cell. (**B**) The correlation analysis of different immune cells. The red color represents positive correlation and the blue color represents negative correlation. (**C**) The violin graph shows the difference in immune infiltration between BA and normal groups. The BA group is shown in blue and normal group is shown in red. *p*-value < 0.05.

**Figure 5 children-09-00697-f005:**
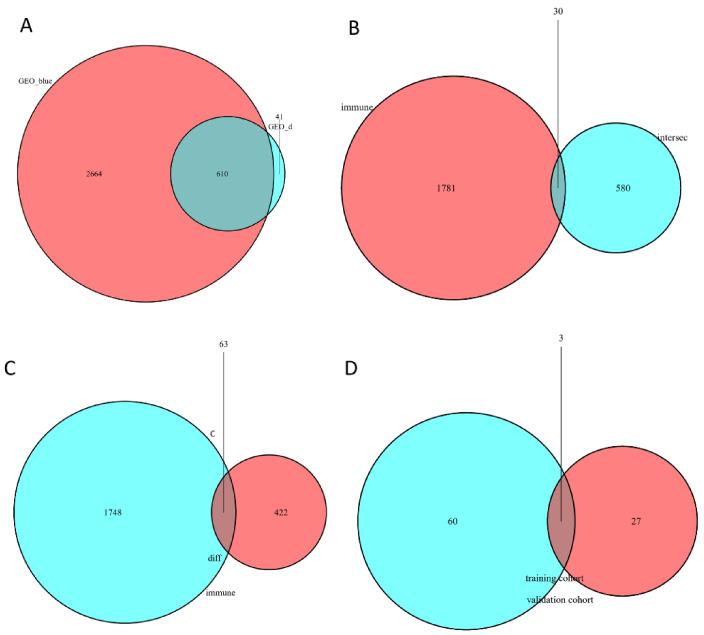
The exploration of BA-related immune genes. (**A**) The overlapping genes of DEGs (bule) and the blue module of WGCNA in GSE122340 (red). (**B**) The intersect genes in GSE122340 of the overlapping genes obtained from the subfigure A (blue) and the immune genes (red) downloaded from database (http://www.immport.org/). (**C**) The intersect genes of DEGs in GSE46960 (red) and the immune genes (blue) downloaded from database (http://www.immport.org/). (**D**) Identification of the target IRGs. Overlapping of immune genes in GSE122340 (red) and GSE46960 (blue).

**Figure 6 children-09-00697-f006:**
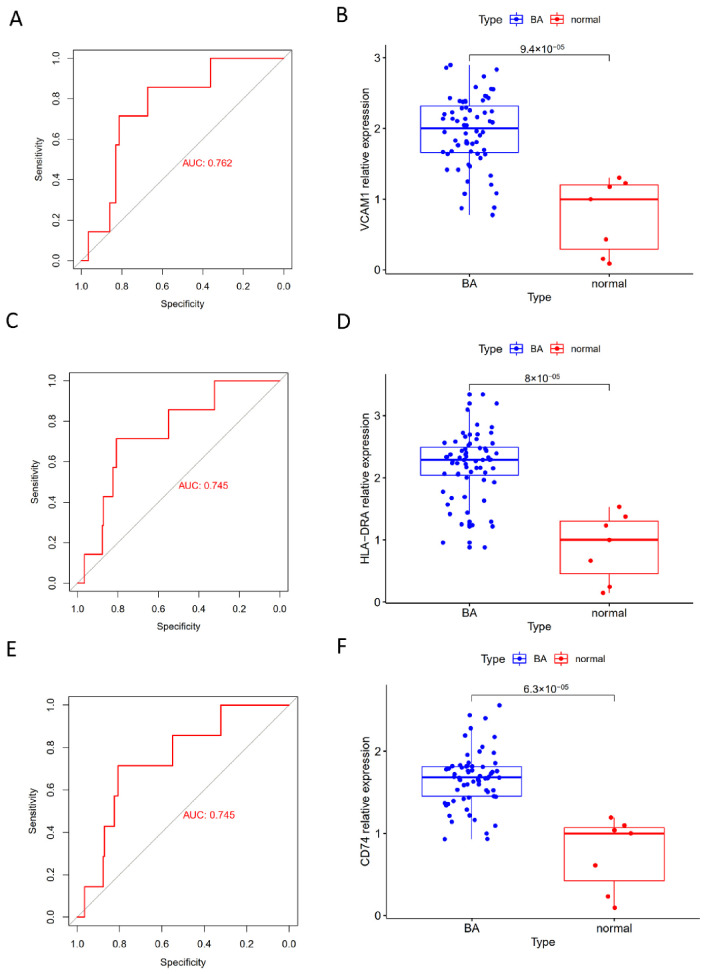
ROC curves of the three hub IRGs in the training cohort and verification of the validation cohort. (**A**) ROC curve of VCAM1 in GSE122340. (**B**) Differences in VCAM1 in GSE46960. (**C**) ROC curve of HLA-DRA in GSE122340. (**D**) Differences in HLA-DRA in GSE46960. (**E**) ROC curve of CD74. (**F**) Differences in CD74 in GSE46960.

**Figure 7 children-09-00697-f007:**
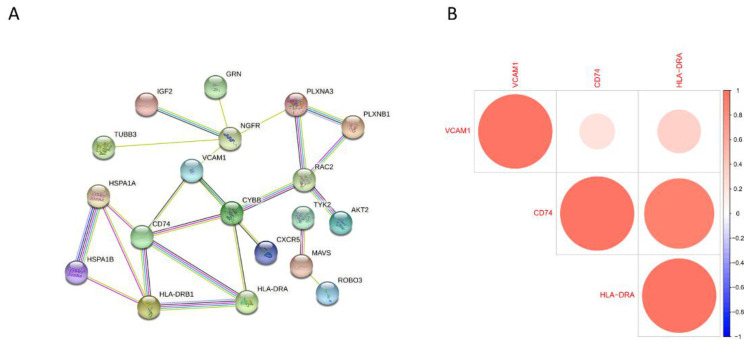
(**A**) The PPI network for the 3 hub genes. (**B**) The interaction between 3 potential biomarkers.

## Data Availability

The data presented in this study are available on request from the corresponding author.

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
