# Peer review of "Identification of Hub Genes and Immune Infiltration in Pediatric Biliary Atresia by Comprehensive Bioinformatics Analysis"

_children, 2022, doi:10.3390/children9050697_

Round 1
Reviewer 1 Report
Congratulations to the authors for their thorough analysis, accurate interpretation of the data, and use of computational analytical methods of WGCNA and CIBERSORT. This manuscript presents useful and important findings that add to the growing understanding of the genetic background of biliary atresia and the importance of immune-related genes to predict survival with the native liver. I did not find any areas that needed correction in this study.
Author Response
Dear professor,
On behalf of all authors,I would like to express our great appreciation to you for comments on our paper .Thank you so much for your approval of our work.
We have made some changes to our manuscript according to another referee. The new version has been uploaded.
Looking forward to hearing from you.
Thank you and best regards,
Yours sincerely,
Yajing Li
Reviewer 2 Report
The authors of this article titled "Identification of hub genes and immune infiltration in pediatric biliary atresia by comprehensive bioinformatics analysis”, investigate genes involved in the immunomodulation related to the biliary atresia, identifying three of them that could serve as potential biomarkers for the condition.
I have read this manuscript with particular interest. The abstract is well structured and offers an organized overview of the study, focusing on objectives, results and conclusions. The introduction is comprehensive and clear. Methods are well explained and comprehensible. the discussion is well structured, clear and understandable, and tackles the debated topics in a comprehensive way. Conclusions are appropriate. The references are complete and up to date. Figures are clear.
Author Response
Dear professor,
On behalf of all authors,I would like to express our great appreciation to you for comments on our paper .Thank you so much for your approval of our work.
We have made some changes to our manuscript according to another referee. The new version has been uploaded.changes are as follows:
1 the introduction part has been reduced.
2 Figure 2 and 3 has been made bigger.
3 A limitation part has been added to the paper.
Looking forward to hearing from you.
Thank you and best regards,
Yours sincerely,
Yajing Li
Reviewer 3 Report
The authors present the outcomes from an experimental study which used gene expression data from the NCBI Gene Expression Omnibus database, used WGCNA to construct key modules and applied CIBERSORT to analyze the immune infiltration in biliary atresia. The authors further explored and verified the novel immune-related genes, with the aim of evaluating if they might serve as potential biomarkers for diagnosis and guide the individualized treatment strategies for BA patients.
This is a very interesting study, which seems well-designed and written. The authors adequately present the published data to date and present their outcomes in an illustrative and clear manner.
A number of issues to be addressed by the authors:
- The introduction section needs to be reduced significantly and some parts could be moved to the discussion section
- Figures 2 and 3 could be made bigger as they are currently hard to read
- A limitations section is indespensable and needs to be added
Author Response
Dear professor,
Thank you so much for giving us the opportunity to revise our manuscript.We appreciate your constructive comments and suggestions.We have modify the paper according your comments carefully .
The introduction part has been reduced.
We are so sorry that the figures are a little hard to see clearly .We have made Figures 2 and 3 as bigger as we can.To tell you a secret, If it is still hard to read ,the figures can be zoomed to get a better experience.
A limitation section has been added.
Looking forward to hear from you .
Yours sincerely,
Yajing
Here are my own thoughts:
I am so surprised and excited to find that my article looks so fresh and concise after deleting some of the introduction part. I appreciate your suggestions so much and many thanks again.